# Optimizing 3D structure of $H_2O$ molecule using DDPG

**Soo Kyung Kim** [* 1] **Peggy Pk Li** [* 1] **Joanne Taery Kim** [2 1] **Piyush Karande** [1] **Yong-Jin T Han** [1]

## Abstract

Conventional methods to predict 3D structure of molecule are based on iterative stochastic optimization techniques based on energy calculation using physics-based electronic structure modeling such as DFT or MD. Therefore, computing cost of physics-based modeling is significantly depended by the number of iterations to calculate energy until the total energy of structure is converged. As the cost-efficient alternatives, we propose a novel RL-based algorithm to optimize 3D structure of single $H_2O$ molecule based on DDPG (Deep Deterministic Policy Gradient) method. To demonstrate the efficiency of our model, we predicted 3D structure of $H_2O$ molecule and compared with results from the conventional DFT calculation. Our experiments show that our model succeed to predict 3D structure of $H_2O$ molecule which is identical with the results from DFT calculation.

## 1. Introduction

In computational chemistry, a common theoretical tool used to determine molecular structure is the geometry optimization procedure. The main idea is optimizing geometry of a given molecular system by minimizing the strain between atoms. Any perturbation from the geometry will induce the system to change, so as to reduce this perturbation unless preventing by external forces. Starting from the experimental geometry of molecule, we calculate total energy of the molecule by slightly perturbing the coordinates of each atom. The calculation of total energy can be done by using simulation methods to calculate electronic structure such as DFT (Density Functional Theory) or MD (Molecular Dynamics). From the variation of total energy, $\delta E(r)$, by chaining location of each atom, $\delta r$, we can estimate the derivative of the energy with respect to the position of the atom, $\delta E/\delta r$. Then, the geometry optimization algorithm

use $E(r)$, $\delta E/\delta r$ and $\delta\delta E/\delta r_i \delta r_j$ to try to minimize the force. There are many geometry optimization techniques which is built on minimizing the strain and the forces on a given system between atoms such as gradient descent, conjugate gradient, or based on Newton's method (BFGS).

There are two main challenges to use this conventional geometry optimization schemes to predict molecular structure.

1. The geometry optimization process seeks to find the geometry of a particular arrangement of the atom perturbed from the initial geometry. Therefore, we can find the local energy minimum nearby initial geometry, but difficult to find a global energy minimum. Also, the choice of the initial coordinate system can be crucial for performing a successful optimization.

2. The process to calculate total energy is repeated until the structure is converged. Therefore, the computing cost of geometry optimization is significantly depended by the number of iterations to calculate energy until the structure is converged. For most large systems of practical interest, it can be prohibitively expensive due to the cost to compute the second derivative of energy.

Here, we propose to use policy gradient reinforcement learning technique as the optimization scheme to predict 3D geometry of molecule. Firstly, as reinforcement learning learns policy by repeating exploration and exploitation, we can potentially explore new structural energy surface which can be dramatically different with initial starting geometry. Therefore, even if we still cannot guarantee whether the output geometry from RL algorithm is the global minimum or not, but the problem that output structure can be stuck on local energy minimum nearby initial geometry *[problem - 1 above]* can be relieved. Secondly, as reinforcement learning seeks to find the best policy to achieve goal rather than find best geometry itself, we can potentially reduce the number of iteration of energy calculation assuming that the agent successfully learn the policy to find the minimum energy in potential energy surface. *[problem - 2 above]* Specifically, to change molecular geometry with continuous movement, we used deterministic policy gradient algorithm (DDPG). To demonstrate the practical use of our model, we show optimization of 3D structure of $H_2O$ molecule.

---

[*]Equal contribution [1]Lawrence Livermore National Laboratories, Livermore, CA, USA [2]Korea University, Seoul, Korea. Correspondence to: Soo Kyung Kim <kim79@llnl.gov>.

*Proceedings of the 36^{th} International Conference on Machine Learning*, Long Beach, CA, USA, PMLR 81, 2019. Copyright 2019 by the author(s).

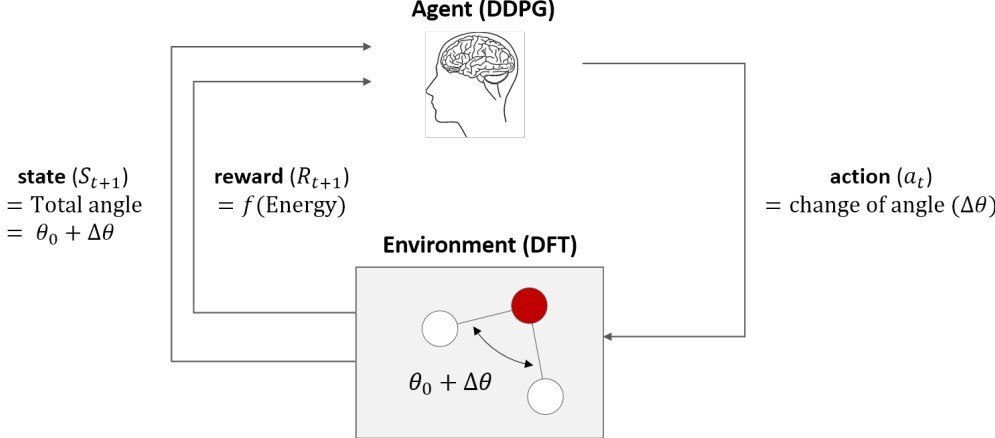

*Figure 1.* Our reinforcement learning setting to optimize structure of $H_2O$ molecule

## 2. Related Work

Recently, with the popularity and development of deep learning, there has been many works in applying deep learning models to predicting molecular attributes and molecular generation (Segler et al., 2017; D Segall, 2012; You et al., 2018; Kusner et al., 2017; Gómez-Bombarelli et al., 2018; Ertl et al., 2017). The most challenging problem in material AI is to find a particularly effective variant of the general deep learning approach and formulate it to chemical or physical domain. Most of advances in material AI built upon graph network representing 2D molecular structure as graph which represent node as atom and edge as the bond between atom. Gilmer et al. (Gilmer et al., 2017) presents Message Passing Neural Network which is adapting gated convolutional neural network to represent molecular structure, and trained model to learn a message passing algorithm and aggregation procedure to predict chemical attribute of molecular graph based on DFT data. However, the idea to apply graphical structure to represent molecular graphs and directly optimize various desired physical property objectives using graphical network can be very challenging. The main difficulty for predicting molecular structure having specific target properties arises because these property objectives are difficult to be featurized (D Segall, 2012) and non-differentiable. Furthermore, the labeled molecular database is significantly limited. As the distribution of the molecules is vast, it is challenging that the supervised neural net based model learns the entire distribution of chemical space to predict meaningful desired properties of specific target material in limited dataset.

As the alternative method to resolve limited labeled data problem of material AI, there have been several advances in applying reinforcement learning to learn chemical or physical properties of molecules. Reinforcement learning based approaches specifically has unique advantages to be applied to molecular prediction or generation task.

1. Desired molecular properties such as drug-likeness (Bickerton et al., 2012; You et al., 2018; Lipinski et al., 1997) and structural attributes such as space group or density (Spek, 2009) are complex and non-differentiable. Therefore, It is difficult to be featurized and directly formulated into the objective function of graph generative models. In contrast, reinforcement learning is capable of directly representing hard constraints and desired properties through the design of state, action and reward function.

2. Second, reinforcement learning allows active exploration of the molecule space beyond samples in a dataset. Therefore, reinforcement learning based approach can be the alternative of supervised deep learning (Hester & Stone, 2013). Specifically, reinforcement learning approach can be the alternative of generative model approaches (Gómez-Bombarelli et al., 2018; Dai et al., 2018; Kusner et al., 2017; Jin et al., 2018). Generative model approaches show promising results on reconstructing molecules which is trained based on given dataset, but their exploration ability is restricted by the limited amount of training dataset.

With above strengths, several goal-directed molecule design models based on reinforcement learning have been proposed recently.

You et al. (You et al., 2018) proposed GCPN (Graph Convolutaional Policy Network) for goal-directed graph generation through reinforcement learning. They trained GCPN model to optimize domain-specific rewards and adversarial loss through policy gradient, and let the trained model acts in an environment that incorporates domain-specific rules. They applied GCPN for designing drug molecule and their results show that GCPN can achieve 61% improvement on chemical property optimization over state-of-the-art base-

lines while resembling known molecules, and achieve 184 % improvement on the constrained property optimization task.

Gabriel et al. (Guimaraes et al., 2017) proposed ORGAN model (Objective Reinforced Generative Adversarial Networks) for generating drug-like SMILES strings. The idea is, in reinforcement learning setting to generate SMILE sequence, they used a sequence based Generative Adversarial Network (GAN) framework (Yu et al., 2017) modeling the data generator as a stochastic policy. Also, they include domain-specifics objective on the top of the discriminator reward. By this combined approach with GAN and reinforcement learning, they showed generation of molecules encoded as text sequences and musical melodies.

Mariya et al. (Popova et al., 2018) proposed reinforcement learning based SMILE generator also using the combined approach with GAN. They first trained generative models with a stack-augmented memory network to produce chemically feasible SMILES strings, and predictive models are derived to forecast the desired properties of the de novo–generated compounds. The generative and predictive models are pre-trained separately with a supervised learning algorithm. Then, both models are trained jointly with the RL approach to bias the generation of new chemical structures toward those with the desired physical properties. By giving reward to have desired properties, their proposed model can be used to design molecule with specific properties. They used their framework to generate SMILE strings of target de novo drug molecule which has a single desired property or multiple desired properties.

## 3. DDPG

**Continuous Action Space :**

Deep-Q-learning (Osband et al., 2016; Mnih et al., 2013; Van Hasselt et al., 2016) shows great success to learn policies in diverse setting, such as Atari game. In DQN, we train function approximator of $Q$ function based on neural nets. If we have function approximator of the optimal action-value function $Q^*(s, a)$, then, in any given state, the optimal action $a^*(s)$ can be found by solving,

$$a^*(s) = \underset{a}{\mathrm{argmax}}\, Q^*(s, a)$$

When there are a finite number of discrete actions, we can easily compute $a^*(s)$ above, as we can just compute the Q-value for each action separately and directly compare them. However when the action space is continuous, DQN cannot exhaustively evaluate the space, and solving the optimization problem above is difficult. Because the action space is continuous, the function $Q^*(s, a)$ is differentiable with respect to the action and we need to solve difficult differential equation.

Instead, DDPG (Lillicrap et al., 2015) algorithm uses an actor-critic approach based on the DPG algorithm to solve above optimization problem.

**Actor-Critic and DPG:**

The Deep-Policy Gradient (DPG) (Silver et al., 2014) scheme contains a parameterized actor function, $\mu(s|\theta^\mu)$. This actor function specifies the current policy by mapping states to action. This actor function is fully deterministic. The another critic function $Q(s, a)$, then, is learned using the Bellman equation as in Q-learning using sampled state and action pairs from actor function. The actor is updated by following equations in a way to maximize expected sum of return from the start distribution $J$ with respect to the actor parameters. DPG proved that this optimization problem is solvable using the policy gradient (Sutton et al., 2000), the gradient of the policy's performance.

$$\bigtriangledown_{\theta^\mu} J \approx \mathbf{E}_{s_t \sim \rho^\beta}[\bigtriangledown_{\theta^\mu} Q(s, a|\theta^Q)|s = s_t, a = \mu(s_t|\theta^\mu)]$$

**DDPG:**

DDPG (Deep Deterministic Policy Gradient) is the modification of DPG which allow it to use neural network function approximators to learn in large state and action space online. There are three main features that DDPG algorithm maintain to improve the performance: (1) Replay Buffer, (2) Soft target update, (3) Manual feature scaling.

First, as in DQN, DDPG used a replay buffer to make the samples are independently and identically distributed. The replay buffer is a finite sized cache **R**. Transitions were sampled from the environment according to the exploration policy ads the tuple $(s_t, a_t, r_t, s_{t+1})$ was stored in the replay buffer. At each timestep the actor and critic are updated by sampling a minibatch uniformly from the buffer

Second, Q-learning algorithm make use of target as below,

$$r + \gamma(1 - d) \max_{a'} Q_\phi(s', a')$$

We try to make the Q-function to be more like this target. Problematically, the target depends on the same parameters we are trying to train: $\phi$. Since $\phi$ being updated by $Q(s, a|\theta^Q)$ is also used in calculating the target value, the Q update is prone to diverge. To resolve this issue, DDPG modify the actor-critic by using "soft" target update, rather than directly copying the weight. DDPG uses target network which make target parameter $\phi_{targ}$ which comes close to $\phi$ but with a time delay. This means that the target values are constrained to change slowly, greatly improving the stability of learning.

Third, when learning from low dimensional feature vector observation, the different components of the observation

may have different physical units. This can make it difficult for the network to learn effectively which generalise across environment with different scales of state values. To resolve this issue, DDPQ manually scale the features so they are in similar ranges across environment and units.

# 4. Methods

We constructed simulated physical environment of $H_2O$ relaxation based on OpenAI gym (Brockman et al., 2016) setting. In this section, we explain details of reinforcement learning setting, and construction of our environment.

## 4.1. Problem setting of Reinforcement Learning

Single water ($H_2O$) molecule has three atoms (two $H$s and one $O$) with two $O - H$ bonds. When optimizing the geometry of $H_2O$ molecule, one aims to obtain the $O - H$ bond lengths and the $H - OH$ bond angle which minimize the total energy. To simplify the problem, we fix the length of $O - H$ bond as theoretical value of 0.96 $\mathring{A}$ and change the $H - OH$ bond angle. We number two $H$ atoms as $H - 1$ and $H - 2$ and only move $H - 2$ atom while $H - 1$ atom is fixed.

Figure 1 shows the overview of reinforcement learning setting of our proposed $H_2O$ relaxation model. First, the agent takes action to change the $H - OH$ bond angle, $\Delta\theta$, from current state of $H_2O$ molecular geometry. Then, according to change of the angle, the molecular geometry is updated and environment evaluate total energy, $E$, from updated molecular geometry. Based on the change of energy, the agent obtain reward. Based on the change of angle, the state is updated. We used DFT calculation as the environment to update molecular geometry and calculate total energy from the given geometry.

Formally, we define the notations as following,

| | |
|---|---|
| $O - H1$ bond length | 0.96 $\mathring{A}$ |
| $H2 - OH1$ bond angle | $\theta$ |
| Energy | $E$ |
| Molecular structure generator | $\boldsymbol{m}$ |
| Energy calculator (DFT) | $\boldsymbol{g}$ |
| Reward function | $\boldsymbol{f}$ |

### 4.1.1. ENERGY CALCULATOR

We initially fixed $x, y, z-$coordinates of $O$ and $H - 1$ atoms with fixed length of $O - H$ bond. As the $\theta$ varies by agent taking the action, the $x, y, z-$coordinates of $H - 2$ atom is updated. The molecular structure generator sub-module, $\boldsymbol{m}$, interactively update the geometry of $H2 - OH1$ molecule.

From the updated geometry by $\boldsymbol{m}$, we calculate total energy using $\boldsymbol{g}$, density functional theory (DFT) calculation. (The description of DFT calculation is followed.) Formally,

$$updated - geometry = \boldsymbol{m}(\theta)$$
$$E = \boldsymbol{g}(\boldsymbol{m}\ (\theta))$$

**Density Functional Theory:**

Density functional theory (Parr, 1980) is the computational method of obtaining an approximate solution to the Shrödinger equation of a many-body system. The quantum mechanical wave-function contains all the information about a given system. For the case of a simple system, we can numerically solve the Schrödinger equation exactly in order to get the wave-function of the system. We can then determine the allowed energy states of the system. Unfortunately due to the multiple variables and difficulties to solve PDE, it is impossible to solve the Schrödinger equation for the system which contains many atoms. Evidently, we involve approximations to render the problem soluble albeit tricky. DFT reduce as far as possible the number of degrees of freedom of the system by the Born-Oppenheimer approximation. In DFT, we approximate Schrödinger equation as the function of the electron density which is a function of space and time. Therefore, we can calculate Schrödinger equation using electronic density approximation. DFT computational codes are used in practise to investigate the structural, magnetic and electronic properties of molecules, materials and defects. Here, we use FHI-aims DFT software-package (Blum et al., 2009) to compute total energy of $H_2O$ molecule given geometry of molecule and pseudo potential with the local-density approximation (LDA) (Stampfl et al., 2001).

### 4.1.2. STATE

The agent should be able to observe the current geometry of $H_2O$ molecule and its energy surface. Therefore, we designed the state as 2 dimensional array with current $H - OH$ bond angle, $\theta$, and current energy, $E$, obtained from the current geometry of $H_2O$ molecule.

We started with the initial $H2 - OH1$ bond angle, $\theta_0$, between 90° to 135°, and let the change of angle, $\Delta\theta$, bounded between -45° to +45°. FHI-aims DFT code terminates the energy calculation when two atoms are located too close each other. We found that the energy calculation using FHI-aims fails for the $\theta$ below 45.0°. To prevent this case, we regularized $\theta$ should be in range between 45° to 180°. If $\theta$ become to below 45° or above 180°, we enforce the $\theta$ should be 45° or 180°.

Before the training, we computed total energy by increasing $\theta$ from 45° to 180° and computed maximum value of energy and minimum value of total energy of $H_2O$ molecule. Figure 2 shows the potential energy surface of $H_2O$ molecule

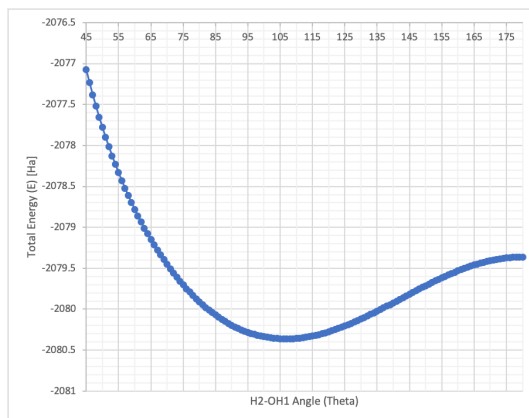

*Figure 2.* Potential Energy Surface of $H_2O$ molecule by changing H2-OH1 angle ($\theta$).

by changing H2-OH1 angle($\theta$). As shown, the total energy is minimum when $\theta$ is between $104°$ to $106°$ . The theoretical $H_2O$ structure has optimal structure at $104.5°$. As shown, the minimum energy is -2080.36 Ha and the maximum energy is -2077.08 Ha. From the computed value of maximum and minimum value of total energy, we normalized the energy, $E$, between 0 and 1.

### 4.1.3. ACTION

In our model, we set the action as one dimensional array representing the change of $H2 - OH1$ bond angle, $\Delta\theta$, from the previous angle, $\theta$. Action is sampled by learned policy in a range between $-45°$ to $+45°$ and the total angle ,$\theta$, is regularized in a way that it is always bounded between $45°$ to $180°$.

### 4.1.4. REWARD

The goal of our model is fining molecular structure with minimum energy. Therefore, we designed reward function, $f$, as the weighted sum of decrease of total energy compared with previous step and decrease of total energy compared with the lowest energy founded so far. Formally,

$$\text{Lowest-energy} = E_{low}$$
$$\text{Current-energy} = E_i$$
$$\text{Previous-energy} = E_{i-1}$$
$$f(E_{low}, E_i, E_{i-1}) = \alpha(E_{i-1} - E_i) + \beta(E_{low} - E_i)$$

As the summary, state, action and reward of our reinforcement learning setting can be represented in table 1.

### 4.2. Algorithm

Algorithm 4.1.2 explains details of DDPG algorithm in our problem setting to optimize 3D structure of $H_2O$ Molecule.

---

**Algorithm 1** Deep Deterministic Policy Gradient Algorithm to optimize 3D structure of $H_2O$ Molecule

1: Input: initial policy parameters $\theta$, Q-function parameters $\phi$, empty replay buffer $\mathcal{D}$
2: Set target parameters equal to main parameters $\theta_{\text{targ}} \leftarrow \theta$, $\phi_{\text{targ}} \leftarrow \phi$
3: $\theta_0 = rand(90, 135)$
4: **repeat**
5:   Observe state (angle $\theta$, energy $E$) $s$ and select action ($\Delta\theta$) $a = \text{clip}(\mu_\theta(s) + \epsilon, a_{Low}, a_{High})$, where $\epsilon \sim \mathcal{N}$, $a_{Low} = -45°$, $a_{High} = +45°$
6:   Execute $a$ in the environment: angle ($\theta$) $= \theta_0 + a$
7:   Observe next state $s'$, reward $r$, and done signal $d$ to indicate whether $s'$ is terminal
8:   Store $(s, a, r, s', d)$ in replay buffer $\mathcal{D}$
9:   If $s'$ is terminal, reset, environment state.
10:   **if** it's time to update **then**
11:     **for** however many updates **do**
12:       Randomly sample a batch of transitions, $B = \{(s, a, r, s', d)\}$ from $\mathcal{D}$
13:       Compute targets

$$y(r, s', d) = r + \gamma(1 - d)Q_{\phi_{\text{targ}}}(s', \mu_{\theta_{\text{targ}}}(s'))$$

14:       Update Q-function by one step of gradient descent using

$$\nabla_\phi \frac{1}{|B|} \sum_{(s,a,r,s',d)\in B} (Q_\phi(s, a) - y(r, s', d))^2$$

15:       Update policy by one step of gradient ascent using

$$\nabla_\theta \frac{1}{|B|} \sum_{s\in B} Q_\phi(s, \mu_\theta(s))$$

16:       Update target networks with

$$\phi_{\text{targ}} \leftarrow \rho\phi_{\text{targ}} + (1 - \rho)\phi$$
$$\theta_{\text{targ}} \leftarrow \rho\theta_{\text{targ}} + (1 - \rho)\theta$$

17:     **end for**
18:   **end if**
19: **until** convergence

---

| State ($s_t$) | $\{\theta_t, \mathrm{E}\}$, where $\theta_t = \theta_{t-1} + \Delta\theta$, $\theta_t = \theta_{t-1} + \Delta\theta$, $\theta \in \{45°, 180°\}$, $E \in \{0, 1\}$ |
|---|---|
| Action ($a_t$) | $\{\Delta\theta\}$, where $\Delta\theta \in \{-45°, 45°\}$ |
| Reward ($r_t$) | $f(E_{low}, E_i, E_{i-1}) = \alpha * (E_{i-1} - E_i) + \beta * (E_{low} - E_i)$ |

*Table 1.* Problem setting of Reinforcement Learning algorithm optimizing 3D structure of $H_2O$ molecule

As the stopping criteria, we stop the searching optimal angle when the change of energy between the energy of the current state with the energy of the previous state is lower than 10e-2 for 10 episodes, and the change of energy between the energy of the current state with the lowest energy found so far is lower than 10e-3.

## 5. Preliminary Experiment

### 5.1. Experimental Settings

We implemented the proposed approach based on the OpenAI Gym (Brockman et al., 2016). We set the learning rate for actor to 0.0001, and the learning rate for critic to 0.001. The batch size is 64, and we used clip normalization.

### 5.2. Experimental Results

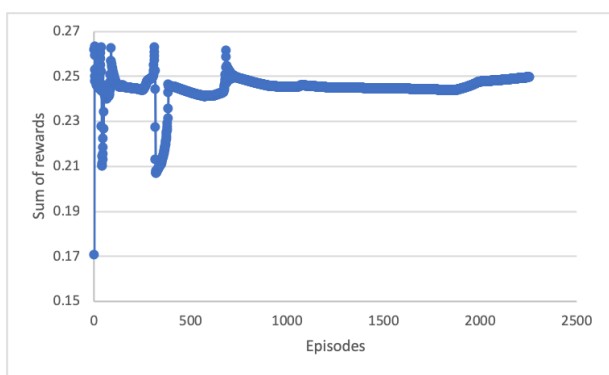

*Figure 3.* Cumulative reward of proposed DDPG molecular structure optimizer

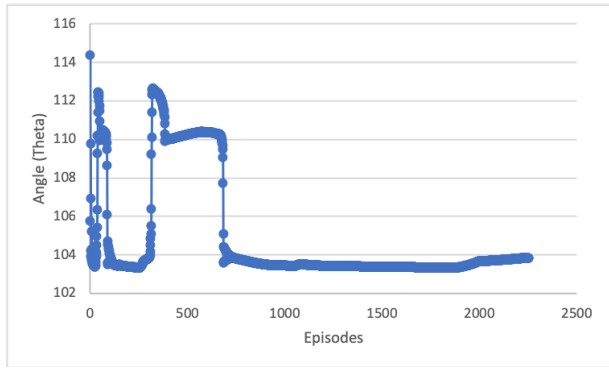

*Figure 4.* Change of angles through episodes

After repeating about 2300 episodes, the proposed algorithm succeed to find the lowest energy configuration which is $\theta$ = 105.95°. The lowest energy we found is slightly different with the theoretical value as 104.5°, but is the lowest energy in our setting with DFT calculation. Figure 3 shows the sum of rewards by episodes until the structure converged, and Figure 4 shows the variation of total angle ($\theta$) through episodes. We see that in the beginning it explores relatively wide area, but gradually converges to the range of true energy (104.5∘) around after 700 episodes. Comparing Figure 3 and 4, we see that the method rewards more when the estimation is closer to the truth.

One issue to discuss is, because we forbid the algorithm to change $H2 - OH1$ angle beyond 180° or below 45°, the actor often stuck on that boundary angles. We consider that the algorithm find the lowest energy fast with removing angle boundary criteria. For this, the change of DFT code itself is required.

## 6. Future Work

Based on this promising results on $H_2O$ experiments, we plan to extend this work in several ways:

- Starting from random points: we currently start from a fixed point to prove the concept. In reality, however, the performance of algorithm is highly affected by the initial state, but we do not have an initial point that is scientifically justified. We may need to try starting from random initial points but still converge to the minimum energy point robustly.

- Relaxation on state restriction: we have limited the space between $45° - 180°$. However, this was arbitrary limitation that we applied as we already know the minimum energy is achieved around $105°$ theoretically. To apply this proposed method for an unknown material, we need to remove these kinds of limitations on the search space.

- Bond-length parametrization: in this paper, we set the bond length to be a constant ($0.96\text{Å}$), but this should be also a parameter to learn. We may add actions to adjust bond length as well as $\theta$ to search best combination of them achieveing lowest energy.

## 7. Conclusion

We propose a DDPG-based algorithm to optimize 3D structure of single $H_2O$ molecule. We performed feasibility study to demonstrate the efficiency of our model by predicting 3D structure of $H_2O$ molecule and compared with results from the conventional DFT calculation. Our experiments shows success example to predict 3D structure of $H_2O$ molecule which is identical with the results from DFT calculation. But convergence is slow and algorithm often stuck on boundary region in action space. To resolve this issue, modification of reward function by changing the DFT code itself is required.

## Acknowledgement

This document was prepared as an account of work sponsored by an agency of the United States government. Neither the United States government nor Lawrence Livermore National Security, LLC, nor any of their employees makes any warranty, expressed or implied, or assumes any legal liability or responsibility for the accuracy, completeness, or usefulness of any information, apparatus, product, or process disclosed, or represents that its use would not infringe privately owned rights. Reference herein to any specific commercial product, process, or service by trade name, trademark, manufacturer, or otherwise does not necessarily constitute or imply its endorsement, recommendation, or favoring by the United States government or Lawrence Livermore National Security, LLC. The views and opinions of authors expressed herein do not necessarily state or reflect those of the United States government or Lawrence Livermore National Security, LLC, and shall not be used for advertising or product endorsement purposes.

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
