# OpenReview forum: "Optimizing 3D structure of H2O molecule using DDPG"
_ICML.cc/2019/Workshop/RL4RealLife — RL4RealLife 2019_

### Official Review · AnonReviewer1 · 2019-05-22
**The paper proposes using an RL-based algorithm (DDPG) to predict the 3D structure of the molecule and the preliminary simulation shows promising results.  I think this work fits well with the workshop.**

**Rating:** 4
**Confidence:** 3

**Review:**

The main challenges of predicting the 3D structure of the molecule and the motivation of why RL-based method is suited for this task are well explained in the paper.   The authors explain in detail how to frame the problem into RL setting (e.g. the design of state and reward).   The simulation result demonstrates that DDPG is able to find the molecular structure of H20, which is a promising preliminary result and also identifies some drawbacks of the current algorithm for future work.

The paper is well written overall. It would be good to have some discussions on how the existing RL-based methods connect with /differ the proposed approach.

---

### Official Review · AnonReviewer2 · 2019-05-27
**Interesting application for computational chemistry**

**Rating:** 4
**Confidence:** 4

**Review:**

The authors employ the Deep Deterministic Policy Gradient algorithm in a proof of concept study to learn to optimize the geometry (by minimizing the energy) of the water molecule, using the molecular geometry as the state and the change of the H-O-H bond angle as the actions, using Density Functional Theory (DFT) as the environment for the energy calculations. The authors are able to reproduce a geometry closely to the one obtained using the traditionally optimization method (e.g. BFGS).

The results are a bit preliminary, but this is nevertheless an interesting approach with lots of potential, which should be investigated further, I therefore vote to accept.

In the related work, I would suggest to acknowledge preceding work on global geometry optimization in computational chemistry (e.g. https://doi.org/10.1002/wcms.70 )

---

### Decision · Program_Chairs · 2019-05-28

Accept